# Automatic Interferogram Selection for SBAS-InSAR Based on Deep Convolutional Neural Networks

**Yufang He [1], Guangzong Zhang [1,\*], Hermann Kaufmann [2] and Guochang Xu [1]**

[1] Institute of Space Science and Applied Technology, Harbin Institute of Technology (Shenzhen), Shenzhen 518055, China; 19b958025@stu.hit.edu.cn (Y.H.); xuguochang@hit.edu.cn (G.X.)

[2] School of Space Science and Physics, Shandong University, Weihai 264209, China; hermann-kaufmann@t-online.de

\* Correspondence: 20b958035@stu.hit.edu.cn

**Abstract:** The small baseline subset of spaceborne interferometric synthetic aperture radar (SBAS-InSAR) technology has become a classical method for monitoring slow deformations through time series analysis with an accuracy in the centimeter or even millimeter range. Thereby, the selection of high-quality interferograms calculated is one of the key operations for the method, since it mainly determines the credibility of the deformation information. Especially in the era of big data, the demand for an automatic and effective selection method of high-quality interferograms in SBAS-InSAR technology is growing. In this paper, a deep convolutional neural network (DCNN) for automatichigh-quality interferogram selection is proposed that provides more efficient image feature extraction capabilities and a better classification performance. Therefore, the ResNet50 (a kind of DCNN) is used to identify and delete interferograms that are severely contaminated. According to simulation experiments and calculated Sentinel-1A data of Shenzhen, China, the proposed approach can significantly separate interferograms affected by turbulences in the atmosphere and by the decorrelation phase. The remarkable performance of the DCNN method is validated by the analysis of the standard deviation of interferograms and the local deformation information compared with the traditional selection method. It is concluded that DCNN algorithms can automatically select high quality interferogram for the SBAS-InSAR method and thus have a significant impact on the precision of surface deformation monitoring.

**Keywords:** SBAS-InSAR; DCNN; ResNet50; automatic interferogram selection

## 1. Introduction

The differential synthetic aperture radar interferometry (D_InSAR) technique has great potential in rapid and large-scale investigations of surface deformations [1]. However, its accuracy is, among others, often limited by long revisit cycles of satellites, low coherences irregular atmospheric turbulence, and the need of external digital elevation models (DEM). Therefore, further techniques and like the small baseline subset (SBAS) method have been developed to exploit sequences of D_InSAR interferograms for monitoring deformation time-series without being significantly affected by decorrelation noise, atmospheric influences and DEM errors affected [2,3]. It can survey the land deformation now with an accuracy in the centimeter or even millimeter range [4,5]. Recently, the InSAR techniques have developed into useful, powerful geodetic surveying tools and are widely applied to landslide monitoring [6,7], mining subsidences [8], surface deformations [9,10], volcanic activities [5], the monitoring of further geological disasters in the context of early warning issues [11,12].

Nowadays, many time series InSAR (TS-InSAR) techniques have been proposed based on the interferograms selection of SBAS strategies, including parallel SBAS (P-SBAS), multidimensional SBAS (M-SBAS) [13], New SBAS (N-SBAS) [14], and pixel-offset SBAS [15]. SAR images contain different complexities and magnitudes of noise, including

those introduced by the atmosphere, the decorrelation process and other sources [16,17], which directly affect the quality of the generated interferograms. Unfortunately, these noise components cannot be precisely determined and separated from the interferograms, especially those induced by the atmosphere and decorrelation process [17–19]. Additionally, the process of calculating interferograms itself directly affects the accuracy of the SBAS-InSAR measurements, whereby the selection of high-quality interferogram pairs is crucial for SBAS data processing. Recently, the selection of interferometric pairs has been improved by various different algorithms. The commonly used method to obtain interferograms is setting optimal temporal and spatial baseline thresholds by exploiting the respective SAR image characteristics [15,20]. However, some low-quality interferograms with large phase errors produced by temporal and spatial variations between two single-look complex (SLC) acquisitions can be mistakenly generated this way. Triangulation reduction and a simulated annealing (SA) searching strategy can improve this constraint in processing data of C-band SAR sensors but needs a long time of operation [21]. The graph theory (GT) can be utilized to delete SAR images and interferograms disturbed by atmospheric turbulences but is not suitable for all SAR images with severe atmospheric distortions [22]. Until now, the most effective approach of high-quality interferogram selection still relies on the traditional manual method. Due to the high degree of human interaction and a large risk of repetitive work, this traditional manual method increases the instability and inconsistency of the deformation calculation.

With an ongoing development of radar satellites and the InSAR technique, the continuous calculation of interferograms becomes increasingly important and thus promotes the development of big data regarding this subject [23]. Nevertheless, the aforementioned common methods of interferogram selection for SBAS-InSAR data processing do not meet the requirements of automation and high efficiency engineering application. In this context, DCNNs, structured by a multi-layer perceptron with multiple hidden layers, have shown their dominant performance on different tasks including image interpretation and classification [24,25]. DCNNs are widely used in the field of computer science, where machines learn specific and complex information from given data by using statistical technologies [26–30]. In geosciences, DCNNs have achieved a superior performance on many applications, such as earthquake magnitude prediction, surface classification, vegetation science, landslide sensitivity mapping, etc. [31–34]. In the past few years, DCNN architecture has adopted transfer learning concepts such as fine tuning and layer freezing, which is superior to the traditional machine learning mode in performance and efficiency to solve the problem of image classification. Considering that the different qualities of interference pairs show different color characteristics, the DCNN method is adopted in this study. Although the training operation of a DCNN model is usually time consuming, it can be used to classify SBAS interferograms with different quality in a very short time.

The residual network is a convolutional neural network model based on residual networks, which won the victory in image classification and object recognition in the Imagenet large scale visual recognition challenge (ilsvrc) in 2015 [35]. The ResNet50 model has the advantages of representing a standard network structure and easy programming. Additionally, it can capture the color change features for the ResNet 50 without the overfitting problem compared to the ResNet10l and the ResNet152 models. Thus, it was chosen to implement the automatic classification of SAR interferograms within the SBAS-InSAR image processing model [35,36]. Section 2 describes the method of the high quality interferogram selection using the ResNet50–DCNN model and the establishment process. In Section 3 the proposed method is applied to the simulated and the actual SAR datasets and its performance is evaluated. Conclusions are presented in Section 4.

## 2. Materials and Methods

In this section, the differential interferogram calculation of the SBAS-InSAR technology and the ResNet50–DCNN model are introduced in detail. The higher the quality of the SBAS-InSAR interferogram, the less intense the color changes within the interferogram

phase in a certain range. This can be used to separate high quality from low quality interferograms by use of the ResNet50–DCNN technique through learning the spatial behavior of color changes within the interferogram phase. The technique has the ability to classify features and learn image priors in the training phase [37–40].

## 2.1. Differential Interferogram Phase of the SBAS-InSAR Technology

The SBAS-InSAR technique was first proposed in 2002 by Berardino et al. [4] which made up for the shortcomings of original PS-InSAR techniques and was mainly used to monitor large-scale deformations. Differential interferograms are formed by setting appropriate parameters of spatial and temporal baselines for SAR images, which can avoid weakening of the relevant factors [3,41,42]. If there are $N$ scene images, M interferograms are formed as follows [4]:

$$\frac{N+1}{2} \leq M \leq N \left( \frac{N+1}{2} \right) \tag{1}$$

A differential interferogram is created from two temporally separated SLC-SAR images and external DEM data. In SBAS-InSAR data processing, a differential interferometric phase can be described by:

$$\Delta \varphi_{int} = \Delta \varphi_{deform} + \Delta \varphi_{dem} + \Delta \varphi_{trop} + \Delta \varphi_{iono} + \Delta \varphi_{noise} \tag{2}$$

$\Delta \varphi_{int}$ denotes the differential interferometric phase, $\Delta \varphi_{dem}$ represents the difference between the actual DEM phase and the applied DEM phase. $\Delta \varphi_{deform}$ is the deformation phase, $\Delta \varphi_{trop}$ and $\Delta \varphi_{iono}$ represent the atmospheric phase variation between two images acquired at different times, and $\Delta \varphi_{noise}$ denotes the noise level (e.g., decorrelation noise phase) [41,42].

$$\Delta \varphi_{others} = \Delta \varphi_{trop} + \Delta \varphi_{iono} + \Delta \varphi_{noise} \tag{3}$$

$$\Delta \varphi_{int} = \Delta \varphi_{deform} + \Delta \varphi_{dem} + \Delta \varphi_{others} \tag{4}$$

In Equation (3), $\Delta \varphi_{others}$ represents other phases including atmospheric phase variations and noise components that need to be separated from the deformation phase during the sequential deformation processing. Thus, the differential interferometric phase is formed by the deformation phase, DEM errors phase and other phases like in Equation (4).

It is noticeable that a differential interferometric phase of high quality has a very low phase change gradient, reflected by minor color changes and a clean map. Thus, phase errors are represented by color changes within the phase map after interferogram unwrapping. Consequently, we designed a classification network using the DCNN technique based on learning the characteristics of the aforementioned phase errors.

## 2.2. Deep Convolution Neural Network

As one of the more famous DCNN working models with very good performance, ResNet is easy to optimize and can improve the accuracy by increasing a considerable depth. We also make experiments to divide similar images into two categories for performance evaluation of different ResNet models including ResNet50, ResNet101 and ResNet152 with the subsequent training sets. When the input size and training times are same set, the accuracy of ResNet50 model is higher than the accuracy of ResNet 101 and 152 models. It is proved that the ResNet50 is better used to capture the color change features and avoid the overfitting problem compared to the ResNet10l and the ResNet152 models. The internal residual block uses jump connection, which alleviates the problem of gradient disappearance caused by increasing the depth in neural network [43,44]. The network is composed of several residual units, which can be regarded as an extension of the two convolution layers. More sufficient feature information can be extracted by adding short connections to the convolution layer [45–47].

Figure 1 shows the structural design of the ResNet50–DCNN model, which is composed of an input layer, convolution layers, identical residual blocks, convolution residual

blocks, pooling layers, a full connection layer and an output layer. The convolutional neural network was constructed by using the Pytorch platform, an open-source deep learning framework developed by the torch7 team of the Facebook Artificial Intelligence Research Institute. Its bottom layer is based on torch, but its implementation and application are all completed by python.

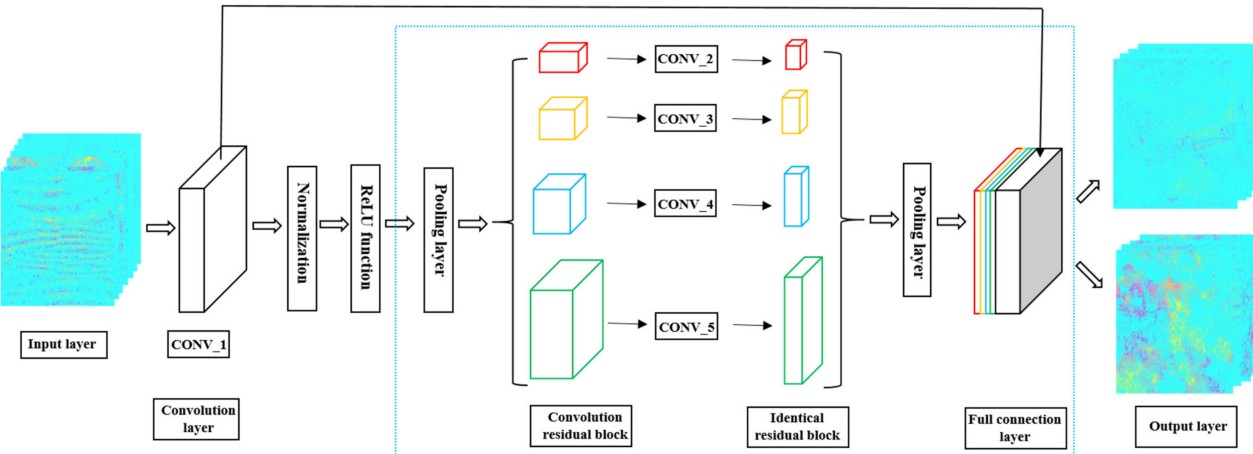

**Figure 1.** Structural design of the ResNet50 model.

A binary classification training model and an appropriate training set have been specifically established for machine learning to automatize interferogram selection and shorten the time of data processing. The input images are sent to the convolution layer, where normalization and an activation function is accomplished. Next, an average pooling is used to reduce the number of samples, and the data is sent to the convolution residual blocks for convolution operation. Then, data are transferred to the identical residual blocks and the remaining identical residual blocks for processing. In order to reduce the amount of data to be calculated, the pooling layer is used to reduce the data dimension after the residual cell. The pooling layer results received are entered into the full connection layer, where the final output results are obtained. Table 1 shows the size of the convolution kernel and the respective output size. The experimental parameters of the real data used in this chapter are described in Section 3.2.

**Table 1.** Convolution kernel sizes and output sizes of the ResNet50 model.

| Layer Name | Output Size | Configuration |
|---|---|---|
| CONV_1 | 1/2 | $7\times7$, 64, stride = 2 |
| CONV_2 | 1/4 | $\begin{bmatrix} 1 \times 1, & 64 \\ 3 \times 3, & 64 \\ 1 \times 1, & 256 \end{bmatrix} \times 3$ |
| CONV_3 | 1/8 | $\begin{bmatrix} 1 \times 1, & 128 \\ 3 \times 3, & 128 \\ 1 \times 1, & 512 \end{bmatrix} \times 4$ |
| CONV_4 | 1/16 | $\begin{bmatrix} 1 \times 1, & 256 \\ 3 \times 3, & 256 \\ 1 \times 1, & 1024 \end{bmatrix} \times 6$ |
| CONV_5 | 1/32 | $\begin{bmatrix} 1 \times 1, & 512 \\ 3 \times 3, & 512 \\ 1 \times 1, & 2048 \end{bmatrix} \times 3$ |
| Classifier | $1\times1$ | Average pooling, Fc (Full connection), 1000, Softmax |

### 2.3. Automatic Interferogram Selection Using the Proposed Method

Figure 2 illustrates the detailed processing flow of the proposed automatic interferogram selection for the SBAS-InSAR algorithm integrated with the ResNet50–DCNN method. In the first step differential interferograms of sequential deformations are calculated from SAR images. Thus, foremost a precise registration between the SAR images is accomplished. After topographic phase removal, the sequential differential interferograms are generated by setting the key parameters of the time and spatial baselines based on the optimal combination conditions and data characteristics. Then differential interferograms are obtained via phase filtering, phase unwrapping and atmospheric correction.

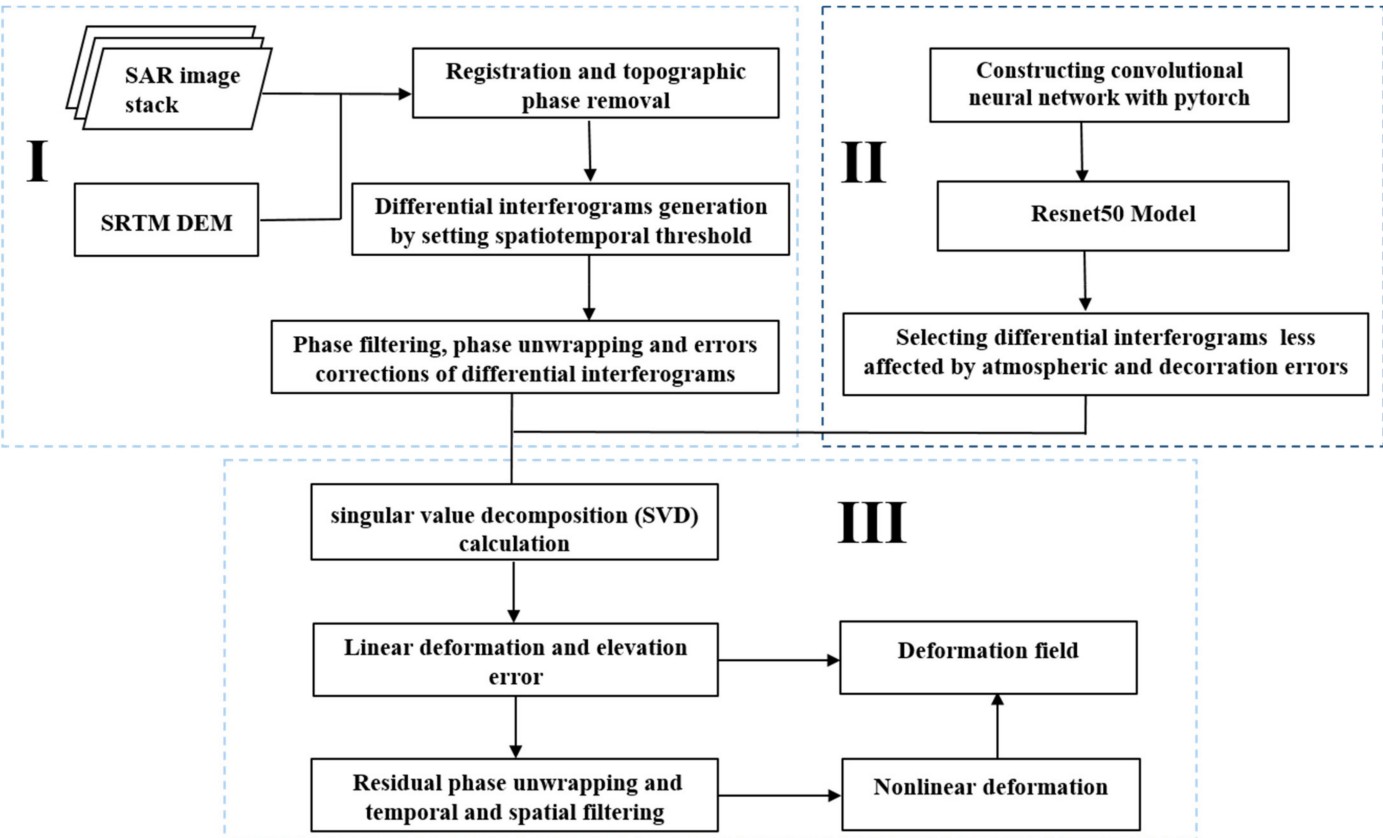

**Figure 2.** Workflow of the proposed automatic interferogram selection for the SBAS-InSAR algorithm integrated with the DCNN method. (**I**) The calculation of differential interferograms of sequential deformations from SAR images. (**II**) The automatic extraction of high-quality interferograms by the ResNet50–DCNN model. (**III**) The estimation of deformation field.

The second part is the most crucial task of the method. The ResNet50–DCNN model is established to realize the automatic extraction of high-quality interferograms. Based on the Pytorch platform, the training model of two classifications for interferograms is constructed, comprising low-quality interferograms with significant errors and high-quality interferogram with only minor errors. Differential interferograms with less errors identified by the ResNet50–DCNN model were finally used to investigate surface deformations of the research area. The third module of the workflow is designed to identify the high coherence points, based on the singular value decomposition (SVD) calculation to estimate the linear deformation and elevation errors. Then, operations of residual phase unwrapping and temporal and spatial filtering were used to separate the nonlinear deformation from the atmospheric delay phase. The sum of the estimated linear and nonlinear deformations represents the entire deformation field. The proposed algorithm efficiently implements

the automatic separation of high-quality and low-quality interferograms needed for the SBAS-InSAR method.

### 2.4. Establishment of Training Sets

For the data processing of the model, it is very important to establish appropriate training sets. Therefore, in this experiment, many phase unwrapping maps of interferograms of surface deformations in different regions with different land deformation characteristics were used. These procedures, including randomly selecting, cutting and filtering phase unwrapped maps, were operated to improve the computational efficiency of the network training [44,48–50]. More than 3000 image patches with different size and varying characteristics of color changes were generated to compose the training sets. Figure 3a,b show a section of the training set of unwrapped interferograms with low and high quality. One can see that the unwrapped interferograms with low quality (Figure 3a) are contaminated by many errors which stand out due to larger color differences through as compared to those interferograms with high quality (Figure 3b).

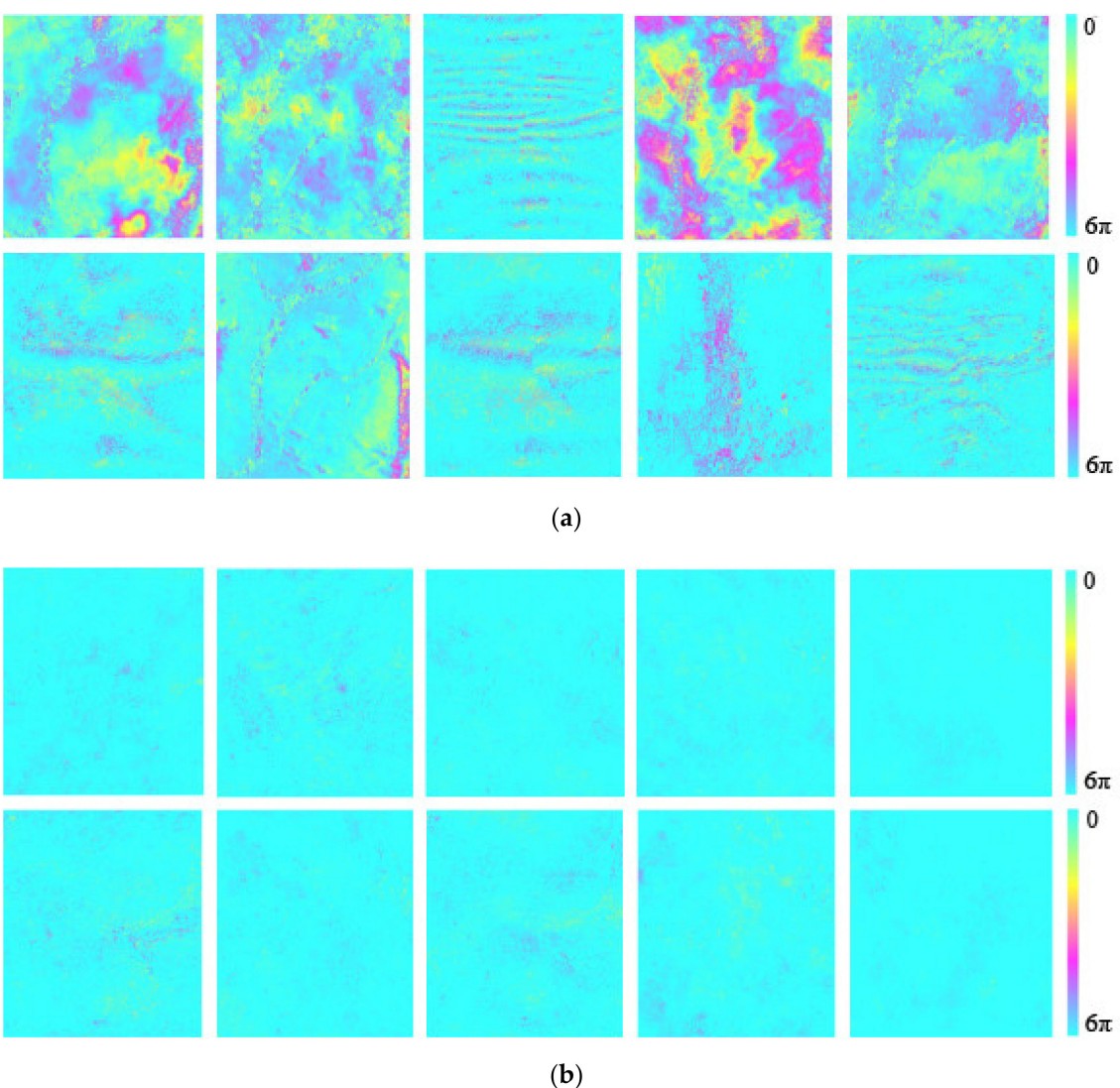

**Figure 3.** Training sets of unwrapped interferograms with (**a**) low quality and (**b**) high quality.

## 3. Results and Discussions

### 3.1. Simulation-Based Tests

A series of simulation experiments were conducted to test the performance of the proposed method for automatic selection of SBAS-InSAR interferograms based on the ResNet50–DCNN model. First, time series of linear deformations were simulated to obtain 222 interferograms. The simulated size of each image is $300 \times 300$ pixels and the subsidence funnel velocity is set to 25 mm per year. Then, 70 interferograms are selected at random to add various turbulent atmospheric errors and decorrelation noise [16,18,51]. Considering the time variability of the turbulent atmosphere, its simulated phase is multiplied by a random constant between 0 and 5 [18]. Here, the decorrelation noise is considered including the time decorrelation error $\Delta\varphi_{dec\_temp}$, a system thermal noise decorrelation $\Delta\varphi_{dec\_thermal}$ and the doppler centroid decorrelation $\Delta\varphi_{dec\_DC}$, which are shown in Equation (5). For the concrete decorrelation error, a polynomial model was established for the doppler centroid decorrelation, and an exponential and a gaussian distribution model were established for a temporary correlation and the system thermal noise decorrelation.

$$\Delta\varphi_{noise} = \Delta\varphi_{dec\_temp} + \Delta\varphi_{dec\_thermal} + \Delta\varphi_{dec\_DC} \tag{5}$$

The temporal decorrelation results from physical changes in the terrain surface such as vegetation growth, soil moisture and other environmental factors between two SAR image acquisitions [52]. Equation (6) shows the temporal correlation ($\gamma_{temp}$), where $\Delta T$ shows the time period according to each interferogram and $\beta$ is a parameter characterizing the temporal decay of the InSAR coherence [53].

$$\gamma_{temp} = \exp(-\beta\Delta T) \tag{6}$$

Figure 4 shows the estimation of the phase standard deviation of the simulated interferograms. It is found that the phase standard deviations of the selected interferograms with atmospheric and decorrelation noise are higher than that of the original interferograms. This indicates the phase standard deviation can reproduce or portray the interferogram quality to a certain degree [22].

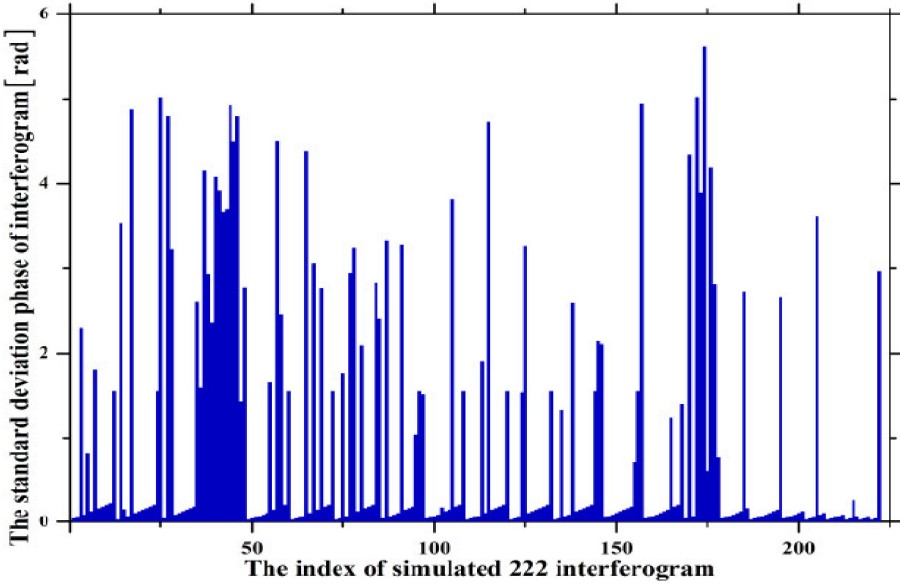

**Figure 4.** Standard deviations of 222 simulated interferograms.

Due to the phenomenon of stronger color change for low quality interferograms, the ResNet50–DCNN architecture can be constructed to eliminate these low quality interferograms. The input images of the network are unwrapped interferograms with error

corrections accomplished. A multi-scale pooling layer is introduced for down-sampling the inputs of the interferogram images. The Rectified Linear Units (ReLu) activation function and the Categorical cross-entropy loss function are applied in the network learning. The back-propagation algorithm of the Adam optimization algorithm is used to initialize and train the ownership value randomly, and the Softmax function is used for classification. Then, the ResNet50 model is established for the training of the binary classification by setting the training cycles to 200, the learning rate to 0.001, and the size of the input image with three RGB channels to 128 × 128. Finally, 168 interferograms with high quality are obtained, and 54 interferograms with low quality are eliminated by the ResNet50 model. Figure 5 shows histogram distribution for standard deviations of the interferogram phase based on interferograms selected by the ResNet50–DCNN method and those originally simulated. It is found that when the interferogram phase standard deviation is lower than 2 rad, the quantity of interferograms obtained by the ResNet50–DCNN method and the simulated interferograms are substantially the same. When the standard deviation of the interferograms ranges between 2 and 10 rad, the quantity of the interferograms is quite different.

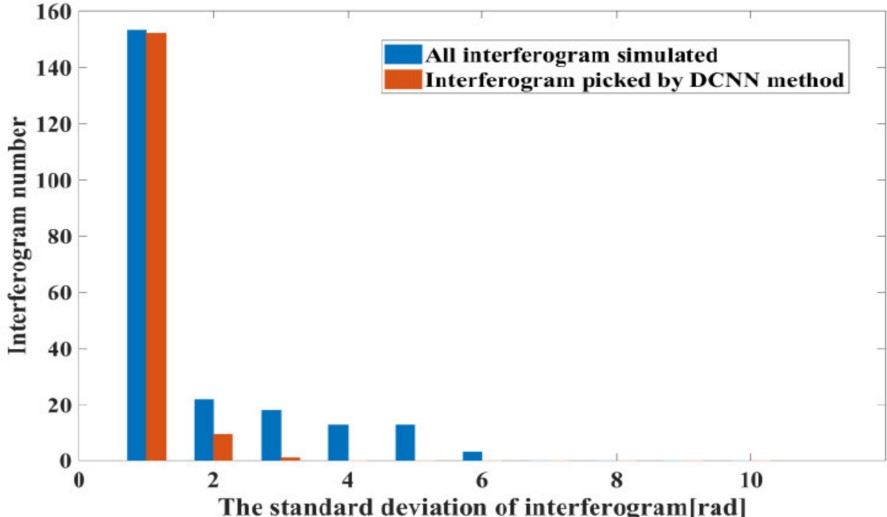

**Figure 5.** Histogram distribution for standard deviations of the interferogram phase based on interferograms selected by the ResNet50–DCNN method and those originally simulated.

Originally, 222 interferograms have been simulated of which 152 were of high quality and 70 of low quality. By comparing the number of high-quality interferograms obtained by the ResNet50–DCNN with the original number of simulated interferograms, it turns out that the accuracy of this method is as high as 90%. Therefore, the ResNet50 model has a significant ability to automatically extract high quality interferograms for the SBAS-InSAR processing.

To obtain the annual deformation rate, stacking technology was applied. Figure 6a displays the estimated annual deformation rates of all interferograms including atmospheric decoherence and decorrelation errors simulated. Figure 6b shows a simulated annual deformation rate of 25 mm per year with no errors and Figure 6c depicts annual deformation rates of interferograms selected by the DCNN method. It is obvious that the annual deformation rate obtained by the ResNet50–DCNN method is in good agreement with the simulations shown in Figure 6b, while the annual deformation rate based on interferograms including all atmospheric and decoherence errors differs quite significantly. The deformation signal in Figure 6a is considerably distorted, and the obviously undeformed areas are contaminated by already described influencing factors. Figure 6d,f provide a somewhat more detailed view of this subject by calculating the differences between 6b and 6a and 6b and 6c, which reflects the advantages of the DCNN based method. Comparing the histograms of Figure 6d, it is noticeable that the one of Figure 6f shows a Gaussian dis-

tribution and the absolute average value is close to 1.5 mm (Figure 6g). Contrary, Figure 6d depicts a non-Gaussian distribution with a double maximum and the average absolute value is close to 4.7 mm (Figure 6e). This further implies the superiority of interferogram selection by the ResNet50–DCNN method.

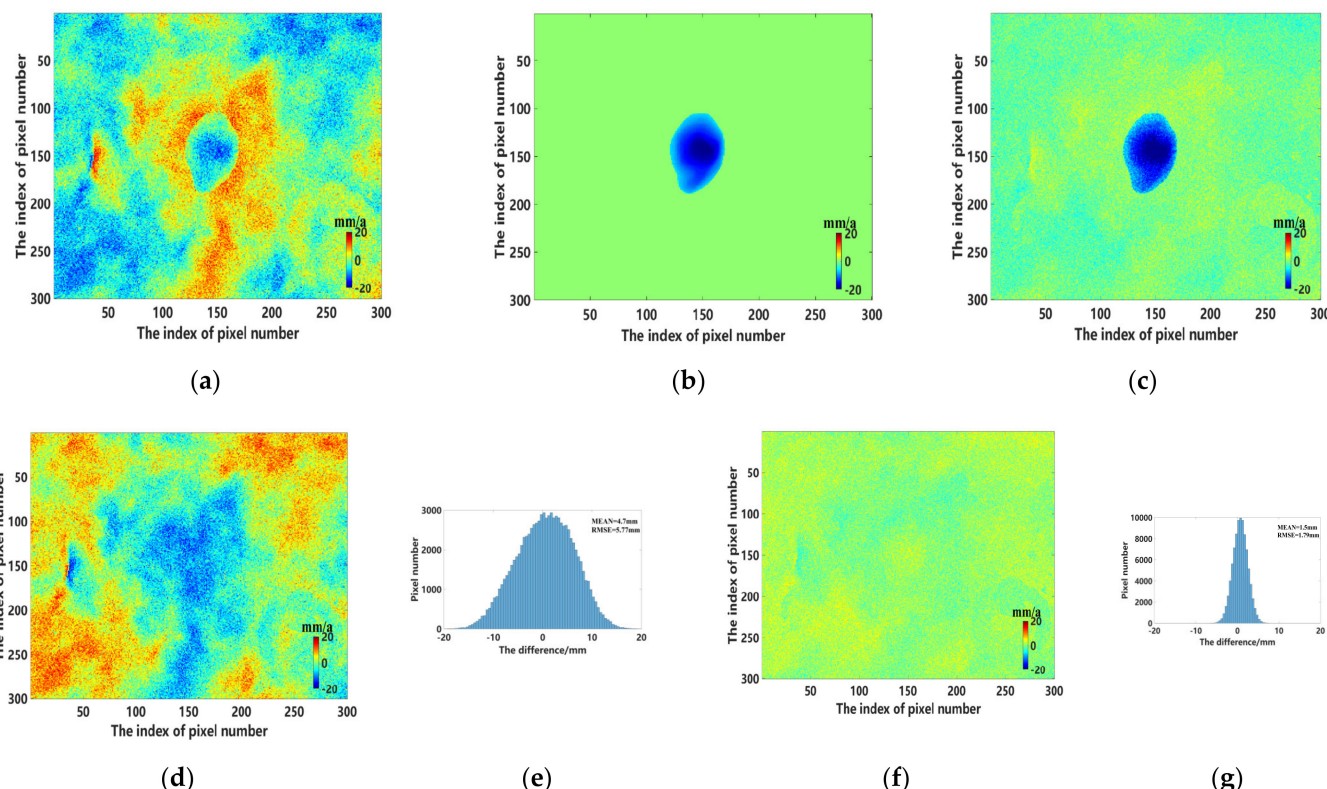

**Figure 6.** The annual deformation rate based on (**a**) the spatio–temporal baseline threshold method (**b**) the manual method (**c**) the ResNet50–DCNN method. (**d**) Difference between (**a**,**b**). (**e**) Histogram of (**d**). (**f**) Differences between (**a**,**c**). (**g**) Histogram of (**f**).

### 3.2. Actual Subsidence Issues

The proposed interferogram selection process based on the DCNN method was finally tested in the Shenzhen area in China (Figure 7), where substantial land subsidences occur due to construction activities at unstable soils in the reclamation areas [54–56]. In this context, a total of 101 Sentinel-1A satellite images taken from March 2017 to December 2020 are applied. Corrections are performed by precise orbits obtained from the European Space Agency (ESA) (https://search.asf.alaska.edu, accessed on 25 October 2021). The reference DEM is obtained from the Shuttle Radar Topography Mission-3 (SRTM) at a resolution of 30 m and used to remove terrain errors in the subsequent differential interference measurements (http://gdex.cr.usgs.gov/gdex, accessed on 25 October 2021). 593 interferograms with a multi-look factor of $4 \times 1$ in range and azimuth directions are generated by setting the time baseline threshold and the space baseline as 75 D, and 150 m respectively. Different training cycles and input sizes of the imagery are set for the ResNet50-DCNN method in order to determine the optimal model parameters. Finally, selected interferograms of the spatio–temporal baseline threshold method and the manual method are used for validation of the proposed DCNN process.

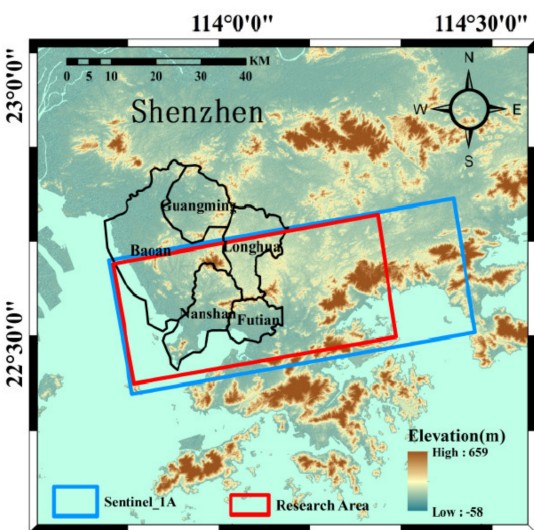

**Figure 7.** Overview of the research area and coverage of SAR datasets.

The application of the ResNet50–DCNN model is seriously influenced by its parameters that have to be determined by analyzing different training results. The training set of two classifications with high-, and low-quality unwrapped interferograms is constructed in Section 3. In addition, the operation of Gaussian adaptive Goldstein is applied to remove noise components, and atmospheric errors are depressed by a polynomial fitting model for the differential interferograms. Finally, the ResNet50–DCNN model was used to classify differential phase unwrapped interferograms. Thereby, the standard deviations of the interferograms are used to analyze the performance of the proposed method [22]. In order to better exploit the model, two key parameters (namely the size of the input image with three RGB channels and the number of training cycles) must be chosen with great care. Table 2 shows the quality of interferograms automatically selected by the model under different parameter settings. The quality is dependent of the interferogram numbers and standard deviations of the interferogram unwrapping phase. The training learning rate was set to 0.001.

**Table 2.** Setting of training parameters and results of the Resnet50 model.

| Size of Input Image | Training Cycles | Interferogram Number | Standard Deviation of Interferogram Phase/Rad | Accuracy (%) |
|---|---|---|---|---|
| 128 × 128 | 100 | 432 | 1.699 | 74.4 |
| 128 × 128 | 200 | 447 | 1.6829 | 86.34 |
| 128 × 128 | 300 | 425 | 1.6406 | 88.53 |
| 128 × 128 | 400 | 461 | 1.7302 | 81.28 |
| 128 × 128 | 500 | 431 | 1.7005 | 80.10 |
| 128 × 128 | 600 | 441 | 1.6749 | 86.00 |
| 128 × 128 | 700 | 464 | 1.7118 | 87.18 |
| 128 × 128 | 800 | 456 | 1.6982 | 85.50 |
| 32 × 32 | 300 | 437 | 1.7537 | 71.0 |
| 64 × 64 | 300 | 448 | 1.7153 | 75.71 |
| 128 × 128 | 300 | 425 | 1.6406 | 88.53 |
| 256 × 256 | 300 | 414 | 1.6225 | 89.00 |
| 512 × 512 | 300 | 411 | 1.6585 | 81.62 |

Figure 8 displays the curves of training efficiency in relation to the number of training cycles within the proposed model. The horizontal axis represents the number of training cycles of the model. The left vertical axis and the right vertical axis represent the standard deviation of the interferogram phase and the percentage of selected interferograms obtained by the model, respectively. The blue pentagram broken line shows the changes for the

standard deviation of the selected interferograms phase at a varying number of training cycles. It is found that the standard deviation (SD) of the selected interferograms is the highest when the training cycles are set to 300. The red rectangle broken line shows the percentage of selected interferograms of high-quality interferograms obtained by manual method at a varying number of training cycles. The ratio increases and then decreases again with increasing training cycles. This behavior confirms that the model has the highest efficiency when the number of training cycles is set to 300.

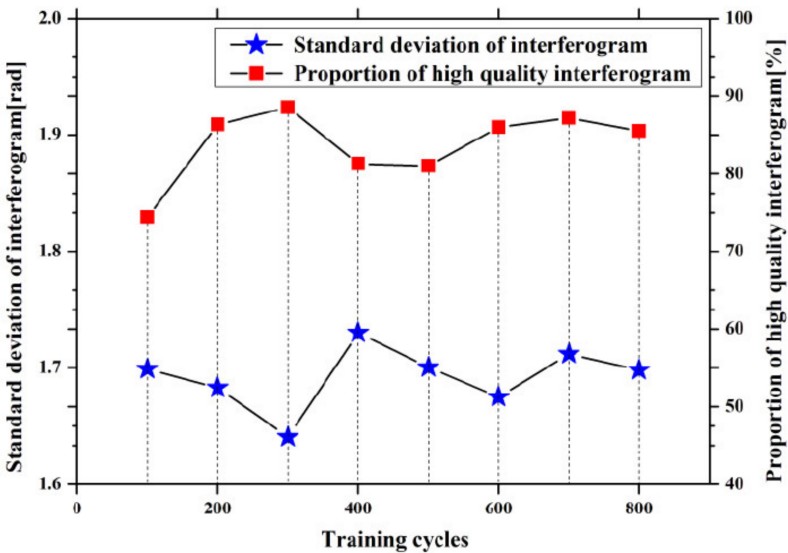

**Figure 8.** Training efficiency in dependency of the number of training cycles.

Figure 9 displays the curve of the training efficiency as it changes with the size of the input image in the training model. The horizontal axis represents the size of the input image with three RGB channels of the Resnet50 model. The blue pentagon polyline shows the phase standard deviation of interferograms selected by the model under different input sizes. The standard deviation phase of the selected interferogram decreases continuously with the size of the input image changing from $32 \times 32$ to $256 \times 256$, and then slightly increases till an input size of $512 \times 512$. The red rectangle broken line also increases and then decreases again with increasing input sizes. It is found that the model has the highest efficiency when the size of the input image is set as $256 \times 256$. Thus, the optimum size of the input image is set to $256 \times 256$ and the number of training cycles is set to 300.

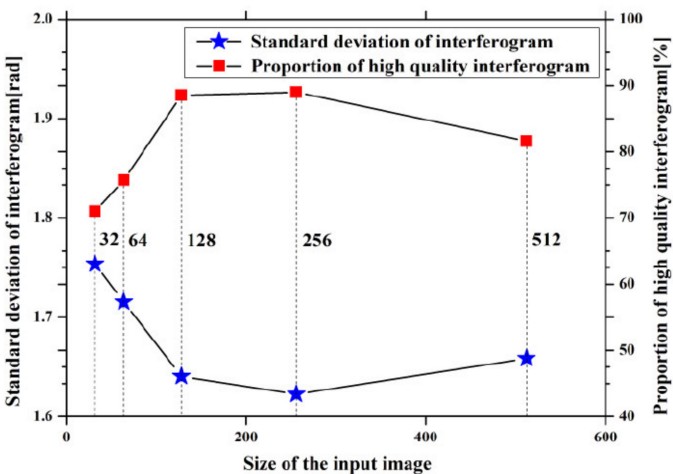

**Figure 9.** Training efficiency in dependency of the size of the input image.

Figure 10a–c display the respective numbers of interferograms extracted by the three different methods. Of course, the interferograms of branch wires obtained based on ResNet50–DCNN method need to be removed. Comparing Figure 10a,b, the two methods deviate significantly, since there are 182 different interferograms and 9 different SLC images. The same is true when collating Figure 10a,c, as there are 168 different interferograms and 9 different SLC images. But there are 14 different interferograms and 0 different SLC images only by comparing Figure 10b,c. Most equal interferograms and SLC images are selected by the manual method and the ResNet50–DCNN method, which can be seen most clearly in areas marked by red dotted line ellipses. It is further indicated that the traditional spatio–temporal baseline method selects some low-quality interferograms, while the ResNet50–DCNN method removes those ones like the manual method.

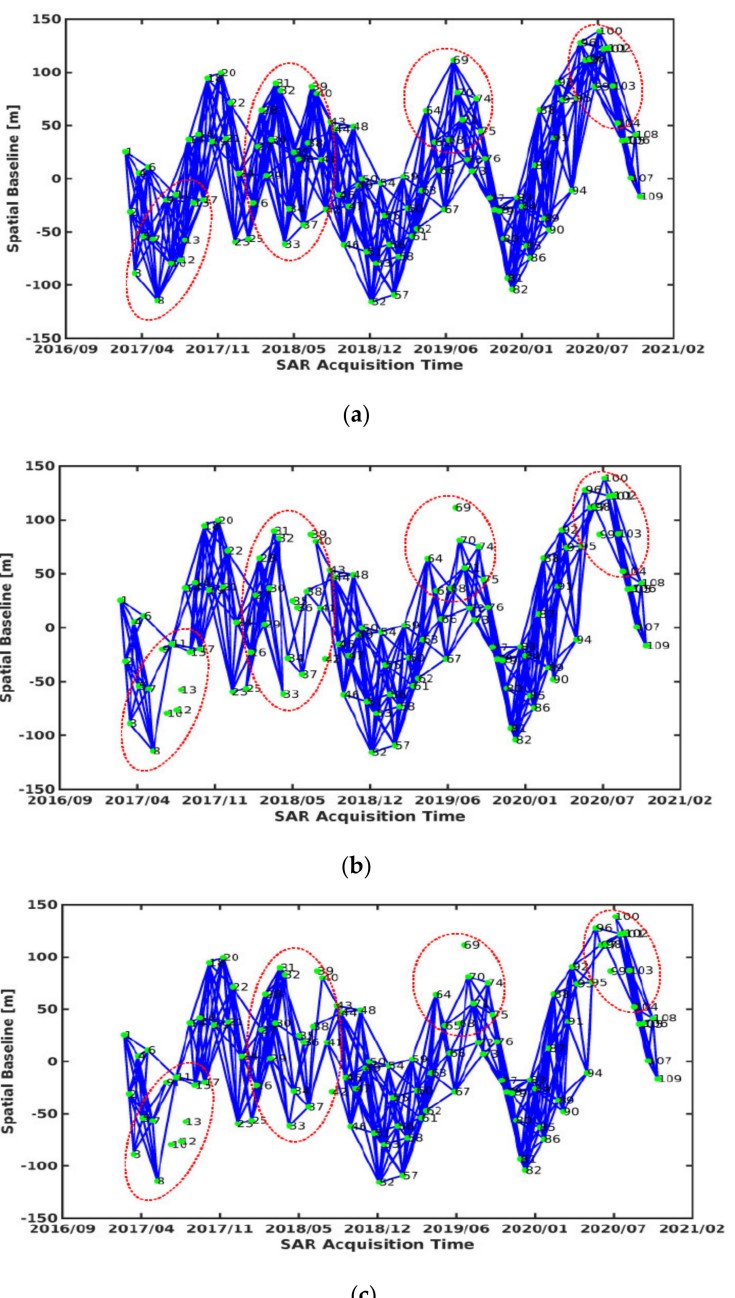

(a)

(b)

(c)

**Figure 10.** Interferograms selected by (**a**) the spatio-temporal baseline threshold method, (**b**) the manual method and (**c**) the ResNet50–DCNN model method.

Table 3 exhibits the retrievable quantity of interferograms and the absolute means and standard deviations of the interferogram unwrapping phase based on the three investigated methods. The number of interferograms obtained by the ResNet50 model nearly matches the number retrieved by the manual method but not the number obtained by the spatio–temporal baseline threshold method. By comparing the number of high-quality interferograms obtained by the ResNet50–DCNN method with the number obtained by the manual method, it turns out that the accuracy of this method is as high as 87%.

**Table 3.** Derived number and standard deviations of the interferogram phase obtained by the three methods.

|  | The Spatio–Temporal Baseline Threshold Method | The Manual Method | The ResNet50–DCNN Method |
|---|---|---|---|
| Number of interferogram | 593 | 411 | 425 |
| Standard deviation of interferogram phase/rad | 2.1054 | 1.6328 | 1.6406 |

In order to confirm the performance of the proposed technique, the quality of all interferograms obtained by the three methods was analyzed and compared in detail. Figure 11 shows the distribution of the standard deviations of the differential interferogram unwrapping phase of three methods by simple histograms. When the standard deviation of the interferogram unwrapping phase is lower than 2.5 rad, the quantity of interferograms obtained by the three methods is basically the same. When the interferogram standard deviation is between 2.5 rad and 5.0 rad, the number of interferograms obtained by the spatio–temporal baseline threshold method is much higher than that of the remaining two methods, indicating that more low-quality interferograms are selected by this method. It is shown that the number of interferograms obtained by the ResNet50–DCNN model is reliable with the smallest overall error of interferograms selected and thus, also reflects the high accuracy of the proposed method depicted in Table 3.

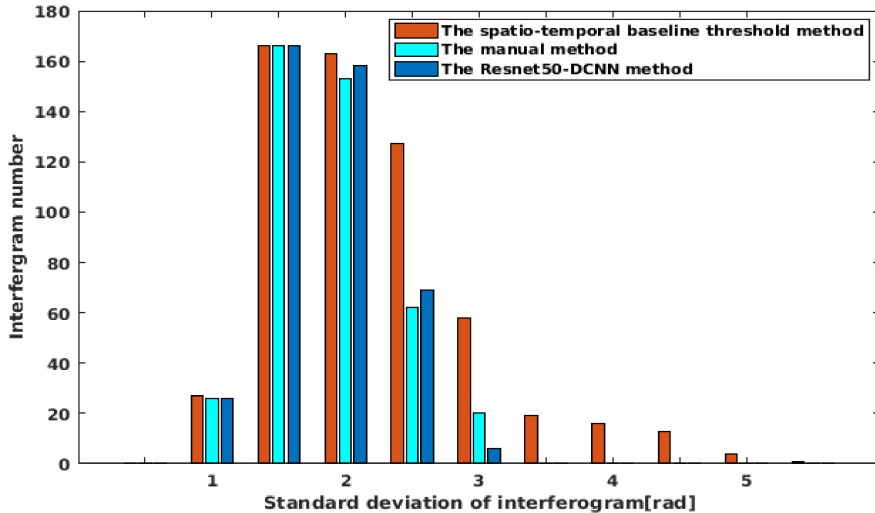

**Figure 11.** Distribution of the standard deviations of the interferogram phase based on the three different methods displayed in form of a histogram.

Figure 12a–c illustrate land subsidence rates in the Shenzhen region based on interferograms selected by the three investigated methods. Figure 12d shows the locations of the 7 PS (point scatters) points (Green circle) and 30 PS points (red triangle) in the Shenzhen area. Figure 12a depicts vertical land deformation rates ranging from −40.15 mm/year to 11.15 mm/year, which are obtained from 593 interferograms based on the spatio–temporal baseline threshold method. It is found that there are different degrees of subsidences around the Baoan airport, the Qianhai Bay, at Houhai and Vanke Cloud City. There is a

large-scale subsidence area along the southeast coast of the Nanshan District and a large-scale land uplift near the Lilin Park. These subsidence rates are contrary to the results derived by other scholars [54,55,57,58]. The results also reveal that interferograms obtained by the spatio–temporal baseline threshold method is strongly influenced by phase unwrapping errors induced by decorrelation noises. Figure 12b shows the vertical deformation rates obtained by the least square method and stacking technology by the manual method. The results show land subsidences ranging from −35.2 mm/year to 14.7 mm/year. The overall land subsidences are relatively stable except from some areas with different degrees of subsidence funnels. The main subsidence areas revealed, are located in the coastal areas, specifically distributed in the C1-C6 zones (Figure 12d) including Bao'an airport, Qianhai Bay, Taizi Bay, Houhai, Zhongxin Mangrove Bay and Vanke Cloud City. This deformation filed is almost the same as the results derived by other scholars [54–58]. Figure 12c depicts the vertical deformation rate of Shenzhen derived by the ResNet50–DCNN model. The land subsidences range from −41.2 mm/year to 15.4 mm/year, and are relatively stable overall, except for distinct places located in the C1-C6 zones. These results roughly agree with the land subsidence characteristics calculated by the manual method and the results of other studies, and confirm the considerable advantages of the ResNet50–DCNN model approach based on automatic extraction of high-quality interferograms.

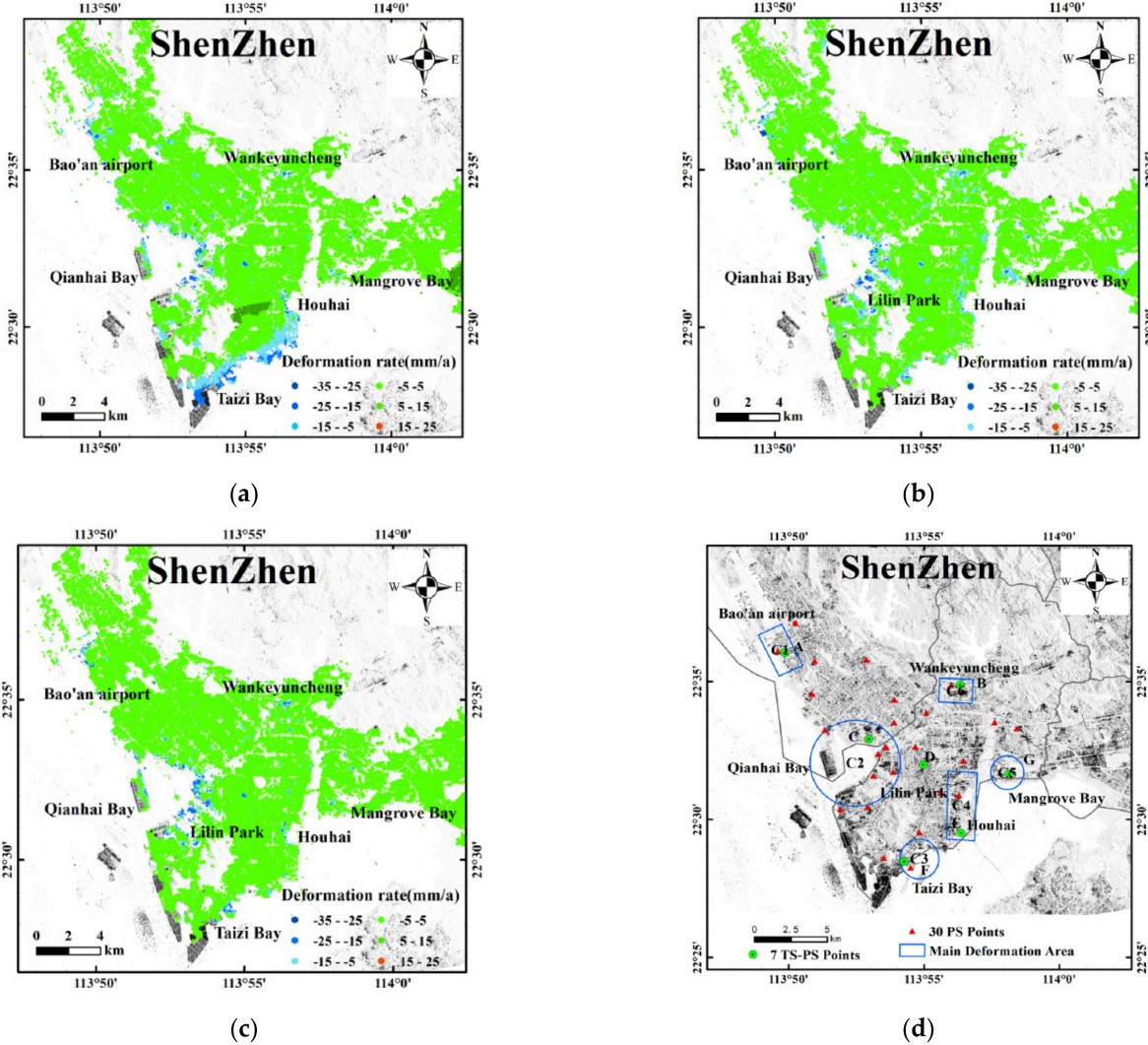

**Figure 12.** Vertical deformation rates based on (**a**) the spatio–temporal baseline threshold method (**b**) the manual method (**c**) the ResNet50-DCNN method. (**d**) Distribution of the main subsidence areas (C1–C6) and the permanent scatter points (PS). Points A–G: Time series of specific PS points (see also Figure 13).

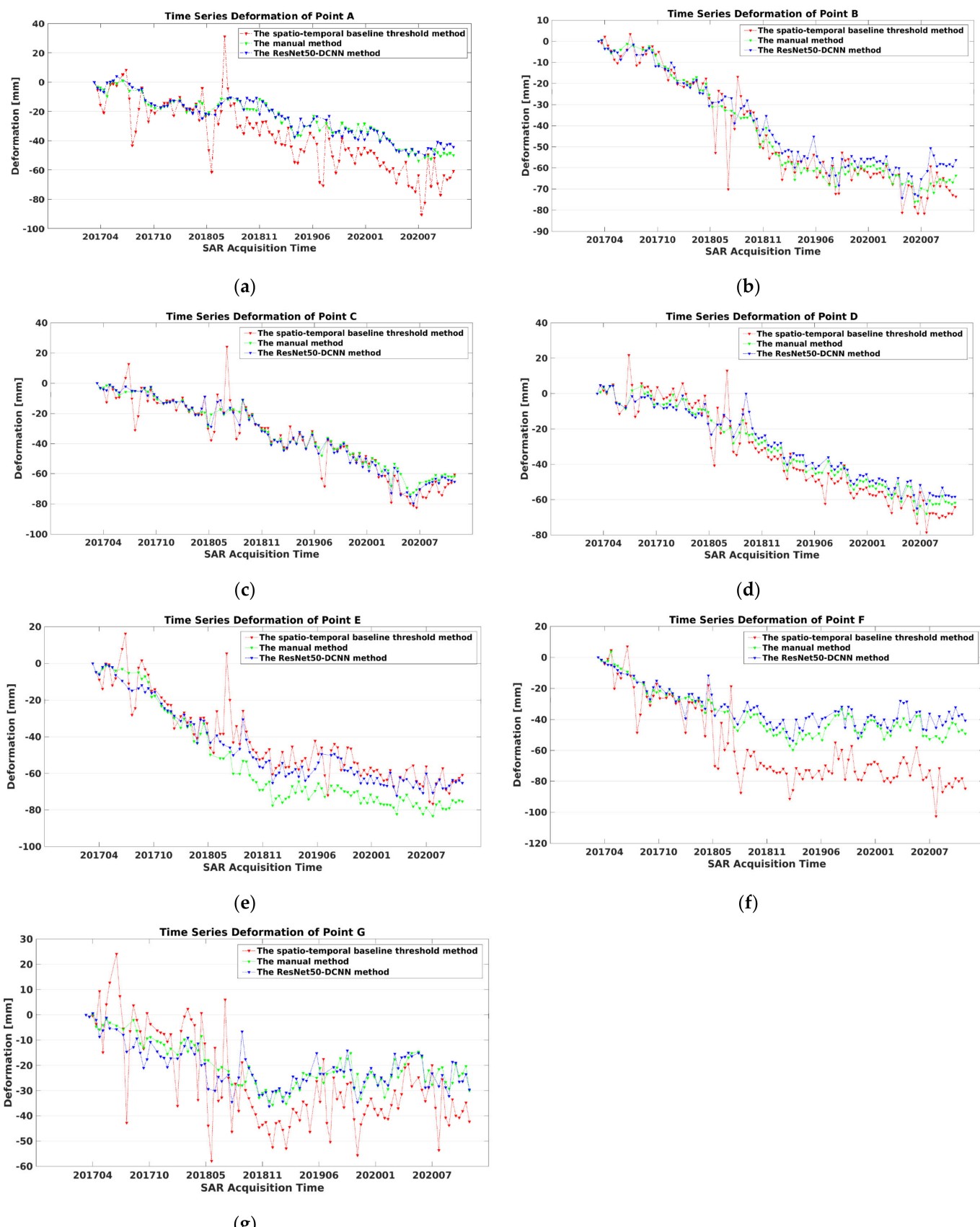

**Figure 13.** Time series of cumulative deformations of 7 PS targets ((**a**–**g**) marked in Figure 12d) based on the three investigated methods.

For further validation of the ResNet50–DCNN model method, time series of cumulative deformations and annual deformation rates of PS were calculated and analyzed. The obtained time series of cumulative deformations of 7 PS targets based on the spatio–temporal baseline threshold method (red triangle dotted line), the detailed manual method (green triangle dotted line) and the proposed method (blue triangle dotted line) are depicted in Figure 13. Data of targets Figure 12a,c,d show a linear regression with continuing subsidence tendency, while targets Figure 12b,e–g display a kind of consolidation of the subsidence trend around the year 2018. Methodologically, it is found that the deflections of single PS measurements are smaller for the ResNet50–DCNN and the manual methods than for the spatio–temporal baseline threshold method. Apparently, the proposed ResNet50–DCNN method performs significantly better than the spatio–temporal baseline threshold method.

Figure 14 shows the annual subsidence rate of 30 evenly distributed and randomly selected permanent scatter points. It is found that the subsidence rates obtained by the manual method largely coincides with that, obtained by the ResNet50–DCNN method, while the rates derived from the spatio–temporal baseline threshold method, exhibit much stronger excursion ranges, especially when the magnitude of the respective subsidence is large.

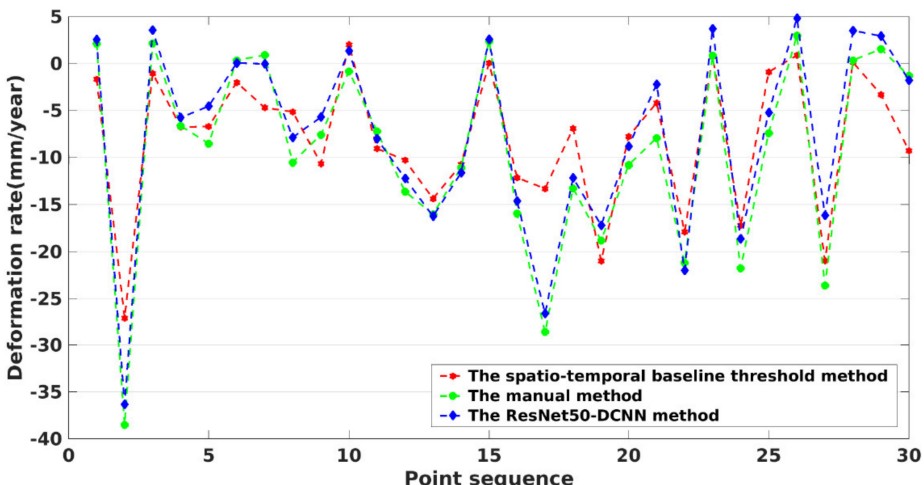

**Figure 14.** Annual average deformation rate of 30 randomly selected PS points obtained by the three investigated methods. See also Figure 12d.

## 4. Conclusions

When monitoring natural disasters such as subsidences, landslides, or mining areas, the selection of suitable uncontaminated interferograms is challenging and time consuming. For this reason, we investigated the possibility and the impact of automatic interferogram selection based on the ResNet50–DCNN model to monitor slow surface subsidences by use of the SBAS-InSAR technique. The idea is based on the fact that interferograms less contaminated by different noise sources display smaller color phase changes within a certain phase range. Hence, training sets containing almost 3000 interferograms obtained from land subsidences in several subregions of Shenzhen in China with varying contaminations of noise were established. Up next, the ResNet50–DCNN model was set up, the respective parameters were determined through analysis of the data sets trained, and traditional interferogram selection methods were used to evaluate the performance. For simulation experiments and the evaluation and validation of real data, phase unwrapping interferograms obtained by the time-spatial baseline threshold method are used to classify high and low quality interferograms based on the ResNet50 model. The quantity of high quality interferograms extracted by the ResNet50–DCNN method is above 90% for the simulation experiment and above 87% concerning the real data experiment, which reflects the accuracy and reliability of the proposed method. A comparison of the overall surface

subsidence rates and the deformation information of local PS points reveals little difference between the land subsidence rates obtained by the ResNet50–DCNN method and the actual simulations or the manual method. The proposed advanced method provides an automatized and fast interferogram selection process for high quality data, which contributes significantly to the application of SBAS-InSAR engineering. For future research, we will expand the training samples and study DCNN models to further improve the general accuracy for a wider applicability of this method.

**Author Contributions:** All authors participated in editing and reviewing the manuscript. Y.H. implemented the methodology, constructed the model, analyzed the InSAR data, produced the results, and wrote the original paper. G.Z. constructed the model and implemented related experiment. H.K. and G.X. supervised the research and revised the manuscript. All authors have read and agreed to the published version of the manuscript.

**Funding:** This work is supported by the Shenzhen Science and Technology Program (Grant No. KQTD20180410161218820) and Guangdong Basic and Applied Basic Research Foundation (No: 2021A1515012600).

**Institutional Review Board Statement:** We choose to exclude this statement because the study did not involve humans or animals.

**Informed Consent Statement:** We choose to exclude this statement because the study did not involve humans.

**Data Availability Statement:** Authors are grateful to the European Space Agency (ESA) for providing the Sentinel-1A SAR data and the precise orbit information free of charge.

**Acknowledgments:** The authors like to thank the anonymous reviewers for their efforts and constructive comments to improve the quality of this paper.

**Conflicts of Interest:** The authors declare no conflict of interest.

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
