# Peer review of "Automatic Interferogram Selection for SBAS-InSAR Based on Deep Convolutional Neural Networks"

_remotesensing, doi:10.3390/rs13214468_

Round 1

Reviewer 1 Report

Now, after this review round the paper has been further improved.

Therefore, I can confirm my recommendation for the publication of the manuscript in its current form.

Reviewer 2 Report

The authors answered all our questions and took into account our recommendations, which significantly improved the manuscript. 

There is just one typo:

The order of the authors of the reference [39] must be:

Arbaoui, A.; Ouahabi, A.; Jacques, S.; Hamiane, M.

Instead of

Hamiane, M.; Jacques, S.; Ouahabi, A.; Arbaoui, A

This manuscript is a resubmission of an earlier submission. The following is a list of the peer review reports and author responses from that submission.

Round 1

Reviewer 1 Report

I don’t think the authors have satisfied all the comments/suggestions. The responses to many of the questions did not show solid argument. clarify what is the 256 bit image size referred to? How come the method is proved to be time efficiency if the training step is not included? Actually the training process is specific to one InSAR data (e.g., Shenzhen), if the dataset is changed (e.g., New York), the training results of the Shenzhen case cannot be used. So even the process of interferogram selection based on the training results is quick (probably not even quicker than the baseline threshold method), the training time should also be considered if you are discussing about the efficiency.

The decorrelation has different physical models, so simulating them should be based on e.g., exponential model for a temporal decorrelation, critical baseline settings for a geometric decorrelation, etc. Also, what is the power of the added noise if the authors use a simple Gaussian. The authors are suggested to present the coherence matrix of the original simulated interferograms, so it’s easier for the readers to understand the designed decorrelations.

The English still has much room to be improved.

Reviewer 2 Report

Dear Editor

The authors have addressed all my comments.

Therefore, I recommend publishing the manuscript in present form.

Reviewer 3 Report

The manuscript presents the studies on an automatic interferogram selection for SBAS-InSAR using deep convolutional neural networks. The text is informatively written, and the overall merit is clearly explained. The authors preceded their studies with a thorough introduction and - from the given version; one can conclude that it presents a corrected resubmission. Hence, the article constitutes a compact form, possibly after solid proofreading, so the current version has evidently been improved. However, the paper contains some shortcomings that need to be refined before publishing. 

Line 13 - Abstract - "millimetre range" is very discursive in terms of  InSAR. I If the authors wanted to leave such a sentence, please provide relevant references further in the text.

The choice of Rasnet50 deep convolutional neural network would need justification (especially beginning from Line 124).

Figure 1 - possible typo in the word "Onput layer".

Figure 3 - the colours of the scale bars are misleading because both -the same colours represent 3Pi and +3Pi. I would recommend changing it. What is more, the differences presented in Part b of the picture are barely visible.

Figure 6 - according to the cartographic principles, the highest values are usually marked red and the lowest blue. Here, it is presented reversely and must be changed.

I believe that after such minor changes, the article could be furtherly processed.

Reviewer 4 Report

This paper proposed  a deep convolutional neural network (DCNN)  to automatically select high-quality interferograms for spaceborne interferometric synthetic aperture radars (SBAS-InSAR).

The manuscript is interesting. However the DCNNs have been used in similar SBAS-InSAR  contests: the Authors must show how this work advances the state of the art with references, discussion, and in particular, with performance comparisons.

Reviewer 5 Report

In this paper, the authors investigated the possibility and the impact of automatic interferogram selection based on the Resnet50-DCNN model to monitor slow surface subsidences by use of the SBAS-InSAR technique. The idea is based on the fact that interferograms less contaminated by different noise sources display smaller color phase changes within a certain phase range

The manuscript is interesting but it absolutely needs an enrichment and an improvement as well methodological as bibliographical opening.

In order to show the importance of Deep learning, it is therefore useful to cite in the introduction some rather diverse applications:

"Deep learning for real-time semantic segmentation: Application in ultrasound imaging. "Pattern Recognit. Lett. 2021144, 27–34.

"An Unsupervised Generative Neural Approach for InSAR Phase Filtering and Coherence Estimation," in IEEE Geoscience and Remote Sensing Letters, doi: 10.1109/LGRS.2020.3010504.

"Past, present, and future of face recognition: A Review. " Electronics 20209, 1188.

"Multi-block color-binarized statistical images for single-sample face recognition. " Sensors 202121, 728.

"Ear recognition based on Deep Unsupervised Active Learning. " IEEE Sens. J. 2021.

Also on page 2, it is important to cite references oriented to ResNet50 applications:

"Concrete Cracks Detection and Monitoring Using Deep Learning-Based Multiresolution Analysis. " Electronics. 2021; 10(15):1772. https://doi.org/10.3390/electronics10151772

Arbaoui, A., Ouahabi, A., Jacques, S. and Hamiane, M. . (2021) “Wavelet-based multiresolution analysis coupled with deep learning to efficiently monitor cracks in concrete”, Frattura ed Integrità Strutturale, 15(58), pp. 33–47. doi: 10.3221/IGF-ESIS.58.03.

In addition, the manuscript has some weaknesses:

- The relationship (1) needs to be justified

- The parameters of the expression (4) are not all defined

- Table 1 has some points to correct: instead of "*", use the multiplication sign, d and fc are not defined.

- In figure 3, part (b) is not clearly explained

- On page 8, line 238, the image size should be 128x128

- On page 12, line 333, we wonder: Is it the pre-trained version of ResNet50 that is used? Why not use ResNet101 or even ResNet152?

- The authors should insist on the possibility of denoising the data as a preprocessing. In this regard, it is possible to take inspiration from wavelet-based denoising and cite the following references:

Ouahabi, A. Signal and Image Multiresolution Analysis; ISTE-Wiley: London, UK; Hoboken, NJ, USA, 2013

"Harmonic propagation of finite amplitude sound beams: Experimental determination of the nonlinearity parameter B/A. " Ultrasonics 200038, 292–296.

"A review of wavelet denoising in medical imaging. " In Proceedings of the 8th International Workshop on Systems, Signal Processing and Their Applications (IEEE/WoSSPA), Algiers, Algeria, 12–15 May 2013; pp. 19–26.

"Nonparametric Denoising Methods Based on Contourlet Transform with Sharp Frequency Localization: Application to Low Exposure Time Electron Microscopy Images. " Entropy 201517, 3461–3478.

The authors say " The quantity of high quality interferograms extracted by the Resnet50-DCNN method is above 90% for the simulation experiment and above 87% concerning the real data experiment, which reflects the accuracy and reliability of the proposed method", how to justify it?